# Current Trends in Organic Vegetable Crop Production: Practices and Techniques

**Juan A. Fernández** [1,2,*]**, Miren Edurne Ayastuy** [3]**, Damián Pablo Belladonna** [3]**, María Micaela Comezaña** [3]**, Josefina Contreras** [1,2]**, Isabel de Maria Mourão** [4]**, Luciano Orden** [3] **and Roberto A. Rodríguez** [3,*]

[1] Departamento de Ingeniería Agronómica, Universidad Politécnica de Cartagena, 30203 Cartagena, Spain
[2] Instituto de Biotecnología Vegetal, Universidad Politécnica de Cartagena, 30202 Cartagena, Spain
[3] Departamento de Agronomía, Universidad Nacional del Sur, Bahía Blanca 8000, Argentina
[4] Escola Superior Agrária de Ponte de Lima, Instituto Politécnico de Viana do Castelo, 4990-706 Ponte de Lima, Portugal
[*] Correspondence: juan.fernandez@upct.es (J.A.F.); rrodrig@uns.edu.ar (R.A.R.)

**Abstract:** Organic farming is a holistic production management system that promotes and enhances agroecosystem health, including biodiversity, biological cycles and soil biological activity, and consequently, it is an efficient and promising approach for sustainable agriculture within a circular and green economy. There has been a rise in the consumption of organic vegetables in the last years because of their organoleptic properties, higher nutritive value and lower risk of chemical residues harmful to health. The recent scientific evidence regarding the use of the major elements responsible for organic vegetable crop production indicates plant material, soil management and crop nutrition, soil disinfection, crop management and pest, disease and weed management. These techniques are the focus of this study. In general, the main outcomes of this review demonstrate that a great effort of innovation and research has been carried out by industry, researchers and farmers in order to reduce the environmental impact of the established and innovative horticultural practices while satisfying the requirements of consumers. However, research-specific studies should be carried out in different farming systems and pedoclimatic conditions to achieve the highest efficiency of these horticultural practices.

**Keywords:** organic production; vegetable; organic crop nutrition; biostimulants; soil disinfection; biological crop management; organic weed management

## 1. Introduction

Organic agriculture, named in some countries as biological or ecological agriculture, is a holistic production management system that promotes and enhances agroecosystem health, including biodiversity, biological cycles, and soil biological activity. It emphasises the use of management practices as preferred to the use of off-farm inputs, taking into account that regional conditions require locally adapted systems [1]. This management system combines tradition, innovation and science to benefit the shared environment and promote fair relationships and good quality of life for all involved [2].

The extent of organically managed farmlands, the number of organic farms, and the global market size for organically grown foods have increased steadily [3]. The latest data show that this tendency was accentuated due to a substantial increase in consumer demand for organic food during the COVID-19 pandemic [4]. Expanding organic production has implied the production of nutritionally improved food crops while using fewer external inputs and reducing environmental impacts [5]. As stated by Willer et al. [4], in 2019, organic agricultural land reached 72.3 million hectares (1.5 percent of the total agricultural land), being managed organically by at least 3.1 million farmers from 187 countries. According to the previous authors, the five countries with the largest areas of

organic land were: Australia (35.7 million hectares), Argentina (3.67 million hectares), Spain (2.35 million hectares), USA (2.33 million hectares) and India (2.30 million hectares), and the distribution of organic agricultural land by continent was: Oceania (50%), Europe (23%), Latin America (12%), Asia (8%), North America (5%) and Africa (3%).

The diets based on organic products seem to be healthier and tastier, providing a better quality of life for people, compared to diets based on conventional foods, although the potential role of the production system has not yet adequately been investigated [6]. Previously published studies on nutritional differences between organic and conventional foods show that the variation in results is very high, depending on several factors such as genotype, fertilisation of plants, ripening stage and plant age at harvest, weather conditions and growing system [7,8]. Therefore, it is not possible to conclude that organic certification can be considered an indication of better overall nutritional quality [9] and that despite the evident toxicological safety guaranteed by organic products, the association between the consumption of organic foods and a reduction in the risk of developing chronic diseases is generally weak [10]. Within organic agriculture, the ecological production of vegetables plays an important role for the sustainable production of organic food and the preservation of the environment. Organic vegetable growers provide healthy food and are a source of livelihood for many farming communities, especially small farmers. Organic horticultural production can improve the quality of life of producers, increase agricultural employment, reduce health risks and improve the sustainability of the agrosystems [11,12].

Organic agriculture is based on management practices that embrace preservation, restoration, maintenance, or enhancement of ecological harmony; it relies on the principles of sustainability, and hence, it helps in attaining objectives of environmental, economic, and social sustainability. Thus, sustainable agriculture emphasises such a production system that can sustain the food needs of all without draining treasured resources. Sustainable agriculture is often referred to as a key system to attain the goal of sustainable development [13].

Organic farming regulations require maintaining and increasing soil fertility and biological activity through multi-year crop rotation; the inclusion of legume crops in the crop rotations as a cover crop or in other green manure crops; soil application of animal manure or other organic matter, preferably both composted and from organic production. To avoid the risks of contamination by nitrates, some regulations such as those in Europe (EU Regulation 2018/848) require that the total amount of organic matter used in organic production may not exceed 170 kg ha$^{-1}$ year$^{-1}$ of organic nitrogen. This limit applies to the use of farmyard manure, dried farmyard and chicken manure, composted farmyard manure and solid animal excrement as chicken manure and liquid animal excrement.

The present review discusses the contemporary state of knowledge on several management practices used in organic vegetable crop production, with an emphasis on current trends in plant material, soil management and crop nutrition, crop management and pests, diseases and weeds management.

## 2. Plant Material

The sustainability of food and agricultural systems depends on seed diversity [14,15], and it is known that wild and traditional cultivars are under enormous human-driven pressures, including industrialised agriculture and climate changes [16].

Currently, organic production mainly relies on cultivars bred in and for conventional production, and only the seeds are organically produced [17], which may not perform as well in organic systems as in conventional systems [18,19]. Orsini et al. reported that the use of organic seed by organic farmers in Europe is low [20], and Knapp and van der Heijden have argued that at least part of the yield gap in organic crop production compared to conventional agriculture might be due to the restricted availability of organic cultivars [19]. However, there are a diversity of landraces that are well-adapted to their local environmental conditions and consumers' preferences, originated through

evolutionary processes that include introgression from wild relatives, hybridisation between cultivars, mutations and natural and human selections [16]. These cultivars are often bred under low-input agricultural conditions and show different levels of genetic diversity, which allows them to cope with the limited use of external inputs in organic agriculture, namely in weeds competition, resistance or tolerance to pests and diseases, nutrient-use efficiency and efficient mineralisation of soil organic matter. Together with crop yield and quality, these are the aims of organic plant breeding [18,21–23].

Many species that are recognised as neglected or underutilised are traditional crops in different regions of the world and important as nutrient supply but unknown in the global market [24]. These species/cultivars could be very important in the context of unpredictable climate change conditions, and some of them were introduced in an organic market that usually values greater diversity, including minor crops [25]. Great efforts can be exemplified by the ongoing projects that boost vegetable organic seed and plant breeding across Europe [26], as well as in the United States [25] and India [27], where studies have shown that organically produced seeds enable better adaptability against climate change, namely better germination under water and salt stress. Further, they support sustainable agriculture through higher nutrient use efficiency and disease resistance. For example, an open-pollinated, early maturing, blocky red bell pepper with broad genetics was released commercially as the 'Renegade Red' pepper, led by the Ecological Farmers Association of Ontario, USA [28].

Organic agriculture has played an important role in the preservation of traditional species and cultivars all over the world and is crucial for sustainable food security, biodiversity conservation and climate change adaptation [29,30]. For example, in Japan, organic farmers use locally available seeds or non-hybrid seeds from seed companies, and many organic farmers save their seeds, aiming to develop their own cultivars adapted to local environmental conditions [31]. A similar approach was reported in other East Asian countries [32]. Across the United States, Canada and Europe, plant breeders, farmers and other stakeholders work jointly to breed new or improved organic crop cultivars in a participatory plant-breeding approach that seems to fill gaps in access to suitable seed, which is not available in the formal seed sector [33,34]. For example, a participatory tomato bred for organic conditions in Italy resulted in an important genetic material for organic agriculture, since these tomatoes assure good productive capacity and seed availability at a low cost [35]. Another example is the breeding of potatoes for organic farming that are resistant to late blight (*Phytophthora infestans*) in a collaborative model of participatory plant breeding in the Netherlands [36]. In participatory breeding, consumers must be involved not only for the input of their taste preferences but also to be acquainted with the importance of agrobiodiversity preservation. In addition, training organic seed producers as well as supporting existing producers is considered essential to increase the organic seed supply.

In Europe, if no organic seed is available, there is a possibility to get derogations for the use of conventionally untreated seed, but the Organic Regulation EU 848/2018 [37] stated that non-organic seed and vegetative planting material are not allowed from 2036 onwards. This regulation also identifies the need to develop cultivars suitable to organic agriculture, namely Organic Heterogeneous Material (OHM) and Organic Varieties suited for organic production (OV). The former, derived from complex crosses between different populations, evolutionary populations (EPs) or farmers' selections, has a high level of phenotypic and genotypic diversity, which allow it to evolve and adapt its phenology to the area of cultivation. As these OHM cultivars are not homogeneous, they cannot pass DUS testing (distinctiveness, uniformity and stability) that is required for the seed market in Europe. However, this obstacle was overcome in 2022, and these seeds can now be marketed after simple notification (delegated regulation (EU) 2021/1189). The OV are derived from organic plant breeding and should comply with the EU Seed Directives. They are suitable to propagate unchanged with stable characteristics of interest over time. To allow for an adequate organic seed market, the European Consortium for Organic

Plant Breeding suggested the integration of traits that are important for organic farming in VCU testing (value of cultivation and use). The EU Green Deal integrates the Farm-to-Fork and Biodiversity strategies [38] and aims for sustainable, climate-neutral food systems, a reduction in pesticides and synthetic fertiliser input and an increase in the organic production area from 9.1% in 2020 [39] to 25% in 2030, signalling the need for fast organic plant breeding and seed production improvement.

Traditional and open pollinated seeds are available in some seed companies in many countries, but the marketing of these seeds by local small-scale seed companies should be developed in order to maintain and distribute these seeds. Bhutan is a good example, as organic agriculture became a national mandate in 2007, and the distribution of seeds for family and market-oriented farmers is ensured by the National Seed Centre [40]. The reduced genetic diversity of the domesticated gene pool has led to the consideration that organic plant breeding should include crop wild relatives, as these are a source of additional genetic diversity that might cope with actual pest and disease pressures, climate changes and market demands [41,42].

The aims of organic plant breeding include obtaining new cultivars resilient to abiotic and biotic stresses associated with organic growing conditions; meeting the market requirements of both quality characteristics and agronomic performance; allowing cultivar reproduction as farm-saved seed; adapting to mixed cropping systems; and taking the best advantage of plant–soil microorganism interactions [18,43].

The grafting of several horticultural crops from the Solanaceae and Cucurbitaceae families on rootstocks of the same species or of another species of the same botanical family has been widely used in conventional horticulture to improve yield and fruit quality under biotic and abiotic stress conditions [44–46]. The interest of vegetable grafting is also centred on the fact that it is an environmentally friendly and easy-to-manage technique that is suitable for organic production. In organic melon production, the cultivars 'Honey Yellow' and 'Arava' grafted onto *Cucumis metulifer* did not increase yield but presented lower galling and reduced root-knot nematode (*Meloidogyne* spp.; RKN) population in the soil, which may be beneficial in double-cropping systems with RKN-susceptible vegetables [47]. For example, in the Netherlands, grafted pepper and cucumber plants are not commonly used in conventional production due to a lack of yield increase, but they are recommended in the organic system to prevent soil-borne disease problems [48]. Grafting green bean with *Phaseolus coccineus* L. rootstocks also appears to be a suitable strategy to increase crop tolerance to the soil-borne fungus disease caused by *Fusarium oxysporum* f.sp. *phaseoli* and to allow a more efficient nutrient uptake, important features for organic production [49].

Grafting that may increase tomato quality under different stress conditions has been reported [50], but results may vary, as, for example, grafting traditional tomato cultivars can induce lower sensory attributes [51]. Similar results were found in Spain, with organic traditional tomato cultivars grafted onto cv. 'Beaufort', that induced negative effects on sensory attributes, reducing sweetness, acidity and the intensity of flavour, and only one grafted cultivar increased yield [52].

The responses of grafted plants to stress conditions depend on the genotype of the graft and rootstock cultivars, the interactions between the two partners and the environmental and soil conditions of the production system. This means that specific studies with different combinations for different conditions are needed, as well as further studies to determine the seedling growth and yield of grafted organic crops. In addition, the progress of a molecular study that clarifies the functions of many genes, proteins and networks of metabolites involved in the responses of grafted plants under different environmental and soil conditions will be important for a more effective selection of rootstocks.

### 3. Organic Crop Nutrition

Worldwide, the organic agriculture regulations do not allow for the utilisation of synthetic products such as fertilisers and industrially manufactured agrochemicals. For this reason, organic farmers use different techniques to increase soil fertility, such as organic amendments, rotations, and cover crops. These agronomic practices are used to improve the biological, chemical, and physical properties of the soil, in addition to supplying nutrients to plants [53].

Fertilisation with animal manure, composted crop residues and leguminous plants as main and intermediate crops are most widely used. There is currently no global regulation on the use of organic fertilisers to be developed all over the world. However, the EU provided a legal framework for organic fertilisers, soil conditioners and nutrients authorised for use in EU organic production, which are specified in the Annex 1 Regulation (EU) 2018/848 [37]. In addition, an adequate crop rotation design should be carried out, including cover crops, green manures and the application of permitted mineral and organic fertilisers [54,55]. These activities produce an increase in the organic matter content of the soil, and the activity of beneficial invertebrates and microbes improves soil physical properties, increases plant nutrient availability, decreases disease risks and increases crop health [54,56].

Likewise, biological nitrogen fixation by legumes instead of chemically synthesised nitrogenous fertilisers is essential for fertility management [55,57]. Poultry, pig, sheep and cattle manure including urine are the main fertilisers of animal origin used in organic production. The use and availability of each is based on the amount available in each region, its price, transportation to the farm and handling [58,59].

Composting is the best strategy for managing organic solid waste. It is based on the bio oxidative decomposition of the original organic materials [60]. Composting transforms organic waste into a humified, stable, odourless, and pathogen-free material that can be used to improve degraded soils [61–63]. Compost is a high-quality product that is used in organic agriculture due to its profitability, respect for the environment, and easy handling [64,65]. Keep in mind that compost provides the crop with a small amount of nutrients, mainly nitrogen, and other materials such as organic or permitted mineral fertilisers are commonly used.

Vermicompost is a product that is increasingly in demand as organic fertiliser in organic farming. It consists of the stabilisation of organic matter by means of earthworms. It has been shown that the content of macronutrients and micronutrients in vermicompost is generally higher than in traditional compost; it contains high levels of the main nutrients in more soluble forms such as nitrogen, phosphorus, potassium, calcium, magnesium and trace elements. In addition, it provides a highly beneficial enzymatic-bacterial microbial load with suppressive and antibiotic action on pathogenic organisms, growth-regulating substances or humic acids that are responsible for plant growth [66]. These properties improve soil fertility physically, chemically and biologically, resulting in higher crop yields.

Another example of organic waste management is biochar, carbon-rich charcoal produced by thermal treatment (pyrolysis) of agricultural residues and organic biomass. It constitutes a sustainable and effective product as a source of organic and mineral nutrients [67,68]. Biochar is used to improve soil health and low soil fertility, raise crop yields, immobilise heavy metals and decrease plant stress. It can retain ammonia, ammonium and nitrates, and it sequesters carbon, contributing to the reduction of global warming [69–72].

Biostimulants and biofertilisers have received increasing interest in the last twenty-five years from the scientific community and the agrochemical industry. Their complementary role in the integrated management of crop nutrition, especially organic crops, has been increasing, positioning itself as an agroecological solution to the problems of fertility, abiotic stress tolerance and quality of organic food production [73,74].

Biostimulants are defined as "*fertilizer products whose function is to stimulate plant nutrition processes regardless of their own nutrient content, with the main objective of improving one or more of the following characteristics: (i) efficiency in the use of nutrients, (ii) tolerance to abiotic stress, (iii) quality, (iv) availability of nutrients confined in the rhizosphere*" [74]. On the other hand, the biofertilisers are living substances, containing living organisms that increase the supply of primary nutrients to the main crop [73].

The benefits of using biostimulants and biofertilisers in horticulture crops are reported, with studies focused on the main scientific knowledge produced in the 2015–2021 period about the use of these technologies around the world. This research was carried out by using Google Scholar for "biostimulants + organic farming" and "biofertilizers + organic farming".

The plant biostimulants include natural substances such as betain, chitin, humic and fulvic acids, vegetal hydrolyzed proteins, phenolic compounds and seaweed extracts (including algae and cyanobacteria) [75].

The biofertilisers that contribute to the growth of plants are: (i) nitrogen-fixing biofertilisers, (ii) phosphate biofertilisers, (iii) biofertilisers for micro-nutrients, (iv) plant growth-promoting rhizobacteria and (v) compost [76].

Regarding the application of biostimulants, we can cite important advances in the study of the use of extracts of macro- and microalgae and cyanobacteria, compounds obtained from crop waste, vegetable hydrolysed proteins and extracts from other plants. Within the biostimulants of marine origin (seaweed), the presence of plant growth regulators (PGRs) with an auxin- and cytokinin-type effect was verified within species of cyanobacteria of the genus *Lithothamiun* in studies on some horticultural crops (*Allium cepa*, *Solanum lycopersicum* and *Vigna radiata*) [77–79]. From crop waste, the obtainment of humic and fulvic acids with the contribution of micro and macro nutrients showed a promising effect in abiotic stress scenarios (salinity, low temperatures) in lettuce crops [80]. Further, the use of extracts from other plants as new sources of biostimulants was studied exploratory, achieving promising results from the phenolic compounds isolated from them [81].

The application of biofertilisers and especially their combination with different forms of organic matter has also been studied recently. The literature cites the positive effects of different consortia of microorganisms with a strong presence of the genus *Azotobacter* [73,80,82] not only in the plant growth and development but also in produce quality [83,84].

Both biostimulants and biofertilisers appear to have a central role in sustainable food production in the future and could be a reliable complement in organic crop nutrition, although an adequate regulatory framework should be developed all over the world. Nevertheless, the new Regulation (EU) no 2019/1009 [85] provides a clear definition of biostimulants linked to their function.

## 4. Soil Disinfection

Worldwide, soil diseases and pests cause difficulties in the management and yield of crops grown in the field or in a greenhouse. The use of synthetic chemical products for soil disinfection is not allowed in organic agriculture, and non-chemical alternatives include biofumigation, solarisation, cultural practices, disinfection of soils with steam, biological control, the application of preventive sanitary measures and innate genetic resistance [86,87].

Biofumigation is based on the action of volatile substances or other chemical or biological processes generated during the decomposition of organic matter for the management of soilborne plant pathogens and weeds [88–90]. The effectiveness of this technique is increased when it is combined with solarisation (biosolarisation) by covering the soil with transparent polyethylene, causing an increase in temperature and retaining volatile biocidal compounds [88,89,91].

The inclusion of organic matter in the soil of vegetal and animal origin that present a C/N ratio between 8–20 has a biofumigant effect. In addition, the incorporation of organic residues can significantly modify the biological, physical and chemical properties of the soil [88], and the procedures for using them may vary depending on the plant species or agroindustrial organic residues, pest or disease to be controlled, quantity and exposure time of the organic material incorporated into the soil [92].

The species of the Brassicaceae family, and within it the genera *Brassica*, *Raphanus*, *Sinapis* and *Eruca*, are the most well-investigated, and their biofumigant effect is because different parts of the plant contain sulphur compounds known as glucosinolates (GSLs). When these compounds are hydrolysed by the enzyme myrosinase, they release volatile compounds, mainly isothiocyanates (ITCs), which have fungal, herbicide, insecticide and nematicide properties [88–90]. These Brassicaceae species can be used as green manure, rotational cultivation, and the incorporation of fresh or dry residues [89,90]. The efficiency of biofumigation with *Brassica* spp. depending on the stage of development, being the pre-flowering phase the most suitable due to the greater accumulation of GSL and, also, biomass fragmentation. The plant residues should be chopped, and the more irregular and larger the pieces, the more heterogeneous the distribution and release of volatile compounds will be [88]. In the greenhouse, the biofumigant efficiency of *Raphanus sativus* and *Eruca sativa* was verified by incorporating them in pieces in plots that were biosolarised for 4 weeks, and results showed a reduction in the population of *Meloidogyne arenaria* and a yield increase of tomato (*Solanum lycopersicum*) plants [91]. Another essential factor in biofumigation or biosolarisation is soil moisture, since water is essential for the hydrolysis of GSLs after cell rupture. The soil must be irrigated up to field capacity immediately after the incorporation of residues to optimise this reaction. Furthermore, as ITCs are volatile, losses can be reduced if the soil is covered with a transparent plastic film after incorporation. The surface is sealed by rolling and/or irrigation in large areas of cultivation [88,90].

Biosolarisation causes a change in the microbial composition of the soil and increases the proportion of antagonists, augmenting the biological control of soilborne diseases and pests. Some bacteria and fungi were very tolerant to ITCs. Among them are different species of *Trichoderma* spp., which are important pathogens' antagonists. The results of an in vitro test confirmed the compatibility of biosolarisation with *Brassica juncea* and *Sinapis alba* on the growth of *Trichoderma* spp. and *Azospirillum brasilense* [93]. The effect of *Brassica juncea* as a biofumigant on bacterial communities showed promising results, because it causes less damage to the quantity and richness of the bacterial community in the soil [94] and has less impact on the process of nitrification and the abundance of microorganisms involved in the soil nitrogen cycle [95]. In the cultivation of strawberries (*Fragaria x ananassa*) in the field or the greenhouse, this species caused an increase in the abundance of arbuscular mycorrhizal fungi and some endophytic taxa, which are beneficial for the roots, favouring plant growth and increasing the yield of commercial fruits [96,97].

Other materials of plant origin (non-brassicas) have insecticidal, nematicidal, and fungicidal properties in their different plant organs, which by soil incorporation exert their biofumigant action. This is the case of *Melia azedarach* (chinaberry) (*Meliaceae*), which possesses limonoid terpenes [87,89]. The degradation of its fruits stimulates the growth of young plants and reduces the severity of pink root disease (*Setophoma terrestris*) in onion crops (*Allium cepa*) [87].

In contrast, *Tagetes* spp. (*Asteraceae*) were more efficient in crop rotations because they present in their tissues the α-terthienyl secondary metabolite, which is found in higher concentrations in the roots [88]. The intercropping of *T. erecta* (African marigold) with tomato proved to be a good management alternative for *Meloidogyne* spp., reducing galls in the susceptible crop and increasing the number and weight of fruits compared to the control without intercropping [86].

Using manure of animal origin as fertiliser is widespread, and soil disinfection generates rapid degradation and high fermentation heat. Poultry manure, especially fresh

waste, has a high nitrogen content, producing ammonium in a greater proportion, a compound associated with the biocidal action of biofumigation [89,90]. In addition, biosolarisation with chicken manure was beneficial for the control of bacterial wilt (*Ralstonia solanacearum*) and increased tomato productivity in the greenhouse and the field [98].

Other organic materials for soil disinfection would be provided by the industrial subproducts. Those from the oil extraction chain result in mustard meals without oil, which were previously incorporated into the tomato plantation, resulting in an easy application and avoiding the work involved in green manure, especially in areas where the frost-free growing season is relatively short [99]. Another would be the chitin or chitosan from the biocomposites industry, which were added to the soil for 6 years causing an increase of 60% in yields in potatoes (*Solanum tuberosum*), carrots (*Daucus carota*), and *Lilium* spp. [100]. The authors argued that the yield increases were not influenced by changes in the chemical properties of the soil, but were a consequence of the variation in the soil microbiota, especially organisms controlling the population of *Pratylenchus penetrans* and *Verticillium dahlia* [100]. Finally, the biosolids or sewage sludge from the processing plants for wastewater depuration present great potential in the suppression of pests and diseases by improving soil quality. Its practical applications as amendments are still strongly limited in field conditions [101].

## 5. Crop Management

A suitable crop management in organic vegetable crops is an alternative to achieve resilient and sustainable agroecosystems. The current production systems use such techniques as mono-cropping, intensive tillage, intensive use of fertilisers and pesticides, etc., which provoke a series of environmental, economic and agronomic issues that can be associated with soil health, greenhouse gas emissions, biodiversity reduction, high production costs, human health, economic risks, etc. Enhancing techniques such as crop diversification, including crop rotation, intercropping, cover crops and enhancement of auxiliary fauna can minimise the negative impacts associated with the abovementioned issues by maintaining an acceptable yield with high food quality and safety in fields under organic horticulture [102]. It should be taken into account that in Europe, there are new rules on organic production, confirming the link with the soil as a basic principle, and as such, the use of "demarcated beds" (greenhouses) is not considered compatible with broader organic principles [103].

### 5.1. Crop Rotation

Crop rotation is a sustainable horticultural practice that consists of growing different species on the same area in sequential growing seasons. In general, organic vegetable farmers are increasingly aware of the benefits of crop diversification, including rotations in their cropping schedules, with the aims of increasing soil fertility, improving soil properties and reducing the incidence of soil-borne diseases [104]. Organic vegetable crops can be cultivated in a wide rotation in open field, since there are numerous crop choices, whereas in organic greenhouse cultivation, the rotations are short, so that a quick build-up of pest populations can occur, causing significant crop damage [105]. In general, as crop diversity enhances through multiple species rotations, significant increases in total nitrogen and soil organic carbon are observed, which improves soil structure and increases the microbial community [106]. Thus, when organic cultivation protocols were associated with optimal crop rotations, the organic yield was only around 10% lower than conventional ones, as opposed to the usual 20% [22].

The inclusion in organic rotations of several leguminous plants considered as vegetable crops, e.g., green beans, fava beans, green peas, cowpeas, etc., increases soil fertility, as such crops are able to form a symbiotic relationship with specific *Rhizobium* bacteria. In this process, Rhizobia fix atmospheric nitrogen and, consequently, there is a lower need for N fertilisers in cropland, and soil biological activity is increased [107].

Other benefits provided by the rotational effects of leguminous plants involve enhancement to soil physical properties, augmentation of soil carbon sequestration, a decrease in the emission of greenhouse gases (GHG), conservation of soil fertility, and breakup of pest and disease life cycles [108], although the efficacy of the abovementioned benefits relies on the leguminous plant that is rotated, particularly in organic vegetable cultivation [109]. However, the enhancement of $N_2O$ emissions and N leaching are some environmental risks from soils rotated with leguminous plants, which can be reduced through a proper rotation system design [110]. In fact, the presence of leguminous plants can raise GHG emissions, particularly $N_2O$, due to N being fixed by the biological nitrogen fixation (BNF) process, although the use of organic fertilisers could further increase such emissions due to the increase in the soil's organic matter mineralisation, so that a thorough assessment for each crop and pedoclimatic condition would be carried out [111].

In addition, as mentioned above, certain species such as *Brassica* spp. crops can be incorporated into an organic vegetable rotation to reduce pest pressure, since they can act as biocidal agents against some pests and diseases [112]. Since the GSL concentration in the cells of the various plants in a family differs substantially, it is crucial to identify species that will be effective in supressing particular soil-borne pests and diseases [113]. In addition, organic farming has led to enhanced concentrations of total GSLs in most of the *Brassica* spp. vegetables and might reveal gene targets that can present resistance against major pests and diseases [114].

*5.2. Intercropping*

Intercropping is becoming more relevant for preserving and improving soil quality and, consequently, crop productivity [115], particularly when it is combined with organic farming [116]. This cultural practice has shown several benefits, including effective nutrient gain, increased microbial diversity, diminished pest, disease and weed damage, and improved land use efficiency [117,118]. Intercropping systems, if well-designed, can lead to a higher combined production per unit area compared with a monoculture (total relative yield and land equivalent ratio >1), although some species can provoke a decrease in productivity, due to allelopathic activity of their root exudates [119]. However, several constrains have caused that intercropping is not widespread in current agriculture, such as the suitability for mechanisation and the request of a single and standardised product [120]. Consequently, it is necessary to properly select the best spatial pattern for each intercropping system, which includes aspects such as machinery and input needs, in order to optimise crop productivity.

Leguminous plants are crucial in numerous intercropping systems, and they are among the most frequently used intercrop species [108]. Intercropping with these plants can help farmers to reduce the amount of fertiliser N, environmental contamination and production costs [121]. Besides BNF by leguminous species, the rise in N availability via intercrops hosting these species occurs because their competitive force for soil N is weaker than that of other plants. Most of the experiences found with intercropping in organic farming deal with the mixture of vegetable crops and different legume cover crops [121,122], but very few of the interactions are between different simultaneous vegetable crops [123–125]. In the latter studies, the use of legumes that fix atmospheric nitrogen as an associated crop has increased the production of the other crops through increases in soil fertility. This is due to the release of fixed nitrogen and the higher microbial activity that can mineralise organic nitrogen. In addition, legumes have an intensive rhizodeposition that activates microbial communities, increasing their diversity and thus soil functionality, with solubilisation of nutrients taking place through the activation of beneficial microorganisms [125,126]. This latter fact is particularly relevant in intercropping with species from the Brassicaceae family, which are not able to develop mycorrhizae and therefore have higher nutritional requirements [127]. Nevertheless, major efforts in research programs on agronomy, physiology and ecology are necessary to contribute simultaneously to the enhancement of intercropping systems; these should

focus mainly on the breeding of intercropping crops, a better knowledge of the plant–microorganism interaction in crop systems, and improved agricultural management and engineering to develop novel, efficient machinery [108].

### 5.3. Cover Crops

Cover crops have been grown for centuries to offer multiple ecological benefits to organic cropping systems, such as improving the physical and biological properties of the soil, avoiding soil erosion, providing nutrients, increasing soil water retention, suppressing weeds, increasing beneficial insects, and reducing soil diseases and nematodes [128]. Particularly in organic farming, cover crops in combination with reduced soil tillage leads to a higher content of soil organic matter [129]. In addition, the incorporation of cover crops into the rotation creates crop diversification, with the advantages that this practice involves. The use of cover crops in organic vegetable production has been analysed by Robacer et al. [130], suggesting some applications for the upcoming development of their management. The authors concluded that the use of cover crops had a series of advantages and potential problems, and that it is necessary to investigate the impact of cover crops under different agro-environmental conditions to obtain more practical conclusions for organic plant producers.

Grasses and leguminous and non-leguminous broadleaves are the main groups of frequently grown cover crops all over the world [131]. The special interest in organic vegetable production is the use of leguminous plants as cover crops, since, as mentioned above, they can be a significant source of nitrogen for subsequent crops due to their ability to fix atmospheric N, although the aforementioned N fixation quantities are dependent on the legume species, growth stage, length of growing season, climate and soil conditions [132]. There are several published studies on the use of leguminous plants as a nitrogen source for the following crops in organic cropping systems. Among the most recent was the experience of Fracchiolla et al. [133] on processing tomato organically grown in Puglia. The authors demonstrated that the use of *Vicia faba* as a cover crop had the highest tomato yield and supplied approximately four-fold higher nitrogen than two other cover crops (*Triticum aestivum* and *Raphanus sativus*). However, Thavarajah et al. [134] studied the effect of cover crops on the nutrient concentration and yield of organic kale, and established that faba bean resulted in the highest prebiotic carbohydrate, protein and mineral concentrations in subsequent kale crops, and ryegrass increased kale biomass production by providing greater N biomass and carbon quantities to subsequent vegetable crops in a shorter time than legume cover crops. Therefore, despite the fact that leguminous crops normally provide higher amounts of nitrogen for subsequent crops, other factors must be taken into account to facilitate these amounts resulting in higher yields.

Additionally, leguminous cover crops can also be intercropped, with a major crop in organic horticulture serving as living mulches to be used for erosion prevention and weed suppression, while achieving high yields and rising the inputs of N via BNF into the whole cropping system [135]. Additional advantages of living mulches systems include the moderation of variations in soil temperature and higher water infiltration [136].

Among the non-leguminous broadleaves, certain species of the Brassica genus can be used as a cover crop in order to decrease pest pressures [137]. Nevertheless, *Brassica* spp. cover crops should be mowed and incorporated to exploit their natural fumigant effect, since the fumigant chemicals are released only when individual plant cells are broken, as mentioned above. Finally, grasses are efficient and productive in converting resources into biomass, although they produce residues with lignin and a high C:N ratio, which can immobilise soil N [138]. Hence, growing mixtures is thought to provide superior ecological services than single species cover crops [139]. However, quantitative information on this subject is nowadays limited [140], and data from large number of experimentations are desirable in order to evaluate the efficacy of those mixtures in different organic cropping systems.

*5.4. Enhancement of Auxiliary Fauna*

The enhancement of auxiliary fauna in organic production systems allows the implementation of conservation biological control, favouring populations of beneficial arthropods and making farming less dependent on external inputs. This practice reduces the great pressure that intensive agriculture has induced on populations of beneficial arthropods, mainly by the negative effects of pesticide use in the agricultural sector [141]. The establishment of habitats of vegetation near to the crops (hedges, flower strips and banker plants) can provide sources of foodstuff (pollen, nectar, arthropods) to a lot of species of auxiliary insects. Consequently, the implementation of these habitats that act as a biological reservoir of auxiliary insects can promote natural pest control, crop pollination and possibly crop production [142]. According to the previous authors, the effectiveness of floral plantings in enhancing crop pollination and pest control services is influenced mainly by plant diversity, the time since establishment and the landscape context. In addition, other aspects must be considered when choosing the floral plantings, such as the flowering period, the number of flowers of the specie, the production of pollen, etc., to obtain a more diversified entomofauna [143]. Some examples of habitats of vegetation near to crops are the study carried out by Alcalá Herrera et al. [144], in which the authors demonstrated that Chrysopidae showed a preference for feeding on *Phacelia tanacetifolia* and *Coriandrum sativum* pollen grains in organic *Brassica oleracea* crops grown in Southern Sweden, as well as that of Kati et al. [145] in central Greece, who showed that *Glebionis coronaria*, *Coriandrum sativum*, *Anethum graveolens* and *Fagopyrum esculentum* can benefit field-grown tomato because were able to attract mainly wild bees species (e.g., *Lasioglossum* spp. and *Hylaeus* spp.), which were the most abundant pollinating insects. In addition, it has been demonstrated that Alyssum (*Lobularia maritima*) selectively attracts and enhances the performance of *Cotesia vestalis*, a parasitoid of *Plutella xylostella* and one of the most important pests in cruciferous plants worldwide [146]. Among the habitats of vegetation, hedges serve as accommodation for the auxiliary insects and as a refuge at the end of the vegetative cycle. They also act as physical protection by attenuating the effects of the wind, erosive processes, etc., mainly in open field cultivation. Regarding flower strips, a higher level of pests and auxiliaries are normally observed in them because the grown plants are more attractive than the crop. Finally, banker plants are used especially in greenhouses, contributing to the development of auxiliary fauna by supplying their necessary resources.

## 6. Pest and Diseases

The fact that synthetic chemical products are not used causes a change in the relative importance of pests and diseases that affect organic crops in relation to conventional ones, as well as in methods of control.

Promoting integrated techniques, including resource conservation mechanisms, and avoiding others that are associated with an intensive consumption of organic inputs would be among the major changes involved in pest and disease control in organic farming.

Many of the techniques used in pest control in organic farming that are common to those of IPM (those that are not chemical) have only a partial effect on pests and diseases and must be combined with other techniques to achieve efficient control.

Currently, pest control approaches in organic agriculture aim to create favourable conditions with healthy plants in a healthy agroecosystem and to apply ecological self-regulation measures in parallel through biodiversity, management measures with a preventive approach and lastly, direct and curative measures [147]. As a starting point, it is very important to have a good base of entomological and pathological knowledge, with a special emphasis on population sampling, in order to be successful in control [148].

### 6.1. Preventive Measure

As mentioned above, the planning of good crop rotation can have beneficial effects in reducing soil pests, diseases and weeds. The main objective is to reduce the incidence of soil-borne diseases by separating the pest from its hosts in time. By alternating campaigns with crops from different botanical families that are not attacked by the same pests or diseases, the cycles of a crop's pests are interrupted, so the pest cannot reproduce due to lack of food. As discussed in the previous sections, Brassicaceae species are interesting for introducing rotation. Planting broccoli before other lettuces not only reduces the incidence of *Sclerotinia minor*, but also the incidence of disease in the next lettuce crop [149]. Further, Larkin and Lynch [150] found disease reduction with Brassica and non-Brassica rotations in potato cropping systems.

Cultural practices that improve crop health include choosing healthy plant varieties that increase tolerance or resistance or reduce susceptibility to pests and diseases. Organic producers must incorporate resistance based on traditional and new plant varieties, but without using GMOs [151].

The availability of resistant commercial varieties is wide and growing continuously; however, it is more common to find organic varieties with resistance to diseases than to pests, since most of them are obtained by molecular techniques. Breeders should select for quantitative resistance because it is durable over time, and they should select cultivars based on both susceptibility to key diseases and pests and positive and negative interactions with minor pests and natural enemies [152].

The use of grafted plants in greenhouses for the control of *Fusarium* spp. in watermelon or tomato, *Verticillium* sp. in eggplant and *Phytopthora* sp. and *Meloidogyne* sp. in pepper is quite common and effective in conventional production; however, the high cost of organically produced scions and rootstocks seeds, as well as all the other inputs (culture medium, fertilisers, pesticides, biological agents, etc.), together with the low demand and lack of production protocols, are drawbacks of development [153].

To reduce the presence of phytopathogens in the soil, solarisation and biofumigation are options that are extensively explained in the previous section of this document. In the bibliography, there are works in which the effectiveness of these techniques is demonstrated [154–159]. These techniques have an effect on *Meloidogyne incognita*, *Rotylenchulus reniformis*, *Pratylenchus* and *Radopholus* nematodes control. Fungi controls to a greater or lesser extent against *Rhizoctonia* sp., *Gaeumannomyces graminis*, *Macrophomina phaseolina*, *Colletothricum gloeosporioides*, *Sclerotium rolfsii*, *Pythium* sp., *Fusarium* sp., *Verticillium* sp. and *Thielaviopsis basicola*, especially when the amendment contains brassicas.

Several studies have shown that the natural resistance of plants to insect pests and diseases is related to the physical, chemical and, above all, biological properties of the soil [160]. The organic matter and microbial activity associated with organic soils provide an optimal balance of nutrients and minerals in plants, which influences the performance of pests [161,162]. Certain specific patterns of gene expression [163] or high levels of accumulation of plant volatiles such as jasmonic acid [164] that are linked to an organic management system seem to increase the defences of the plants and to be related to the natural control of pests.

Physical methods, such as the use of row covers, coloured plastic and conventional mulches, are a promising strategy for preventing pest and insect vectors from reaching crops. Among the main pests that can be prevented are whiteflies (*Bemisia tabaci* and *Trialeurodes vaporiarorum*), thrips (*Frankliniella occidentalis*), aphids (*Myzus persicae*, *Aphis gossipii* and others) and leaf miners (*Liriomyza* spp.). Physical barriers also reduce the incidence of the viruses that these vectors transmit. As for mulches, some research has shown that they can reduce insect pest pressure. Thus, whitefly (*Bemisia argentifolii*) colonisation was hampered in zucchini crops when wheat straw and reflective mulches were used [165].

Plant biodiversity management is the basic strategy that enhances natural enemy impact and exerts direct effect on pest populations. There are many studies linking increased biodiversity with a positive effect on biological pest control [166]. The agroecological theory on complex food chains leads to a greater diversity of plants affecting a greater diversity of herbivores, and in turn promotes a greater diversity of parasites and predators [167]. On the one hand, pest populations are reduced because pests have difficulty finding their host plant, since they cause interference in the behaviour of the pest in search of the host. On the other hand, there is an increase in natural enemies because these plants serve as source of food, shelter and laying substrate.

As previously mentioned, rotations, intercropping and polycultures are ways of increasing biodiversity in time and space within the crop plot. Another way is through the installation of ecological infrastructures in the field of cultivation (hedges, flower strips or banker plants). Increased biodiversity encourages conservation biological control; however, sometimes the interactions between the individuals that make up the agroecosystem make the results of biological control unpredictable and can condition success, giving rise to additive or negative effects on pest populations [168]. Therefore, much more research is needed to provide recommendations to growers to ensure consistent and predictable pest control.

### 6.2. Curative Measures

Pheromones are a powerful tool to regulate arthropod populations allowed in organic farming. The use of semiochemicals for monitoring pests is quite useful when populations are low, but it is less effective in cases of massive infestations [169]. The disruption of pheromone mating is successfully used for lepidoptera and as a mass trapping method for lepidoptera and thrips; however, these techniques are not applicable to other taxa. Investigations into the use of sex pheromones in other taxa should be encouraged for future organic pest management [148].

Mass release of beneficial organisms can be useful when there are not enough natural enemies or when they arrive too late. This practice is more common in greenhouses than in the outdoors. For timely applications, data on pest population field dynamics by monitoring are required [148].

In terms of chemical control, most pesticides are prohibited by organic standards. However, active substances contained in plant protection products authorised for use in EU organic production are specified in the Annex 1 Regulation (EU) 2018/848) cited above [37]. The current trend is that the use of some of the permitted chemicals (i.e., copper) is being limited or even prohibited to avoid the risk to the health of people and the environment. Therefore, maintaining the efficacy of the few pesticides used in organic production requires using other non-chemical means as a first line of defence [170]. If, despite prevention efforts, pest problems occur, chemicals should be used but only as a last resort. Chemical pesticides must be chosen to prevent not target damage. A full understanding of natural pesticides' compatibility with natural enemies and pollinators is necessary, as well as a greater knowledge of the effect of pesticides on the health of ecosystems, plants, animals, and humans.

Organic pesticides have lower toxicities and fewer residue issues than conventional ones, so they are considered less effective. This results in more applications that have negative effects, such as the appearance of resistances.

When plants are attacked by herbivorous arthropods and pathogens, they activate their defense mechanisms to repel these attacks by emitting volatile organic compounds called HIPVS (herbivore-induced plant volatiles). Advances in HIPVS knowledge have opened up a new field of research for enhancing the effectiveness of host plant resistance and biological control for pest management [171].

The fact that the approval of active substances is based on the registration criteria for synthetic substances creates technical difficulties in adapting them to natural substances. Therefore, many useful products in the protection of organic crops are marketed as

biostimulants under the law that governs fertilisers, which differs between countries. Others have been legally introduced on the market without registration as a result of a legal loophole. In principle, their composition is unknown, which is a reasonable cause for concern [172].

It would be necessary to establish a category of natural substances with its own regulations to solve at least some of these problems.

## 7. Organic Weed Management

Weed management in organic systems is a complex process [173,174] that requires flexibility, knowledge of weed biology and management strategies [175]. Weed management strategies in these production systems include preventive, cultural, physical, mechanical, thermal and biological methods [174].

According to Melander et al. [176], preventive and cultural weed management are organised around three objectives: reducing weed density, reducing interference by surviving weeds against crops and preventing undesirable shifts in weed population and community composition. Different principles and their associated practices are used to meet these guidelines: disturbance niche elimination biocontrol (crop rotation, cover cropping, tillage, living and non-living mulching and seed predation), competition (crop spacing and density, soil fertility, intercropping and cultivar choice) and diversity (the use of many and diverse practices, a diverse crop rotation and cover cropping, identification, monitoring and management of weed flora and direct weed control) [174,176–180].

Physical methods include manual and mechanical weed management. In organic vegetable farming systems and smallholder agriculture, hand-weeding is one of the oldest and most effective methods of weed management between plants and rows, especially for less competitive crops [174,178,181,182]. So far, weed management in organic agriculture has been dependent on tillage. Though this method can be effective in controlling weeds, it can negatively affect soil conservation. As an alternative, advances in precision agriculture for weed management (the use of spot tillage, strip tillage and inter-row cultivation) can be incorporated to maintain some of the soil conservation advantages of no tillage [183]. In this context, mechanical weed management methods have been developed, with substantial improvements for inter-row weed management (e.g., hoeing, split-hoeing, brush-hoeing). These methods are generally effective and assure high crop selectivity, and they are widely used in vegetable crops [176,181]. Currently, there seems to be a trend toward the use of high biomass cover crop residue for weed management in organic systems [183,184], which must be complemented with techniques such as decapitating weeds when they are flowering, suppressing them by mowing or rolling [178].

Thermal weed management is obtained when thermal energy is transferred to plant material in a manner that causes the plant structures to denature and eventually die. Some of these methods, such as solarisation, flaming and steaming, are currently used in organic weed control [176,181,185,186].

Bioherbicides are biological control agents that are applied in similar ways to chemical herbicides as to weed management. Potential bioherbicides may be developed from pathogens (most commonly, the micro-organism used is a fungus) [174], or they may be plant-based, such as natural by-products, allelochemicals and extracts of natural material [187]. Some essential oil extracts, evaluated in pots and field assays, have shown a promising bioherbicide potential [188,189]. Since essential oils can decrease the growth parameters of weeds by reducing their fitness and competitiveness, allowing the crop to outcompete them, another advantage of their use could be maintaining high biodiversity by not completely eradicating weeds [189]. Furthermore, the use of allelochemical compounds in plants with allelopathic potential such as the Brassica species can be applied through different strategies for weed management. These strategies include application as an aqueous extract, soil incorporation, biofumigation, mulching, intercropping, cover cropping and inclusion as a crop in the rotation [184,190].

In a review of the ecological approaches for weed science future, the authors emphasise four key ecological principles: increasing diversity (at different scales), reducing resource availability to weeds, using 'little hammers, not sledgehammers' (the use of multiple management techniques instead of techniques with high selection pressure) and taking advantage of the positive effects of weeds [191]. These approaches prioritise the integrity and resilience of the agroecosystem in large spatial and temporal scales, using these as a base for designing sustainable systems and weed management strategies in a given field and growing season. This techniques also promote multiple synergies with other agroecosystem functions (such as pollinator support and pest management) [191,192]. Transdisciplinary work, scientists' and rural extensionists' engagement in participatory approaches and practices and the consideration of local farmer opinions and constraints have been recognised as keys to future weed science and will be of fundamental importance to the adoption of ecological weed management approaches and methods [192,193].

## 8. Conclusions

All over the world, the use of traditional species and cultivars in organic agriculture has played a crucial role in food and agricultural systems sustainability, food security, biodiversity conservation and climate change adaptation. Current scientific studies on the use of practices and techniques in organic vegetable crop production are presented in this review, highlighting the high level of research performed on plant material, soil management and crop nutrition, soil disinfection, crop management and pest, disease and weed management.

Currently, cultivars suitable for organic agriculture, either organic heterogeneous plant material or organic cultivars suited for organic production focused on resiliency to abiotic and biotic stresses, are being developed worldwide. In addition, the use of grafted vegetable plants for their resistance or tolerance to abiotic and biotic stresses needs a more effective selection of rootstocks for the various environmental and soil conditions of production systems.

Fertilisation with animal manure, composted crop residues and leguminous plants as main and catch crops are usually the main strategies used for organic crop nutrition. However, the use of biofertilisers and biostimulants is becoming increasingly important in the sector, because these are able to stimulate plant nutrition processes to improve nutrient use efficiency, tolerance to abiotic stress, quality and availability of nutrients confined to the rhizosphere, whereby the most commonly used products are based on algae extracts, humic and fulvic acids obtained from crop waste, plant extracts and functional microorganisms.

Traditional agricultural techniques such as crop rotation, intercropping and the of cover crops can help to offset excess carbon dioxide in the atmosphere and thus fight climate change. The above techniques together with the current increase in the use of auxiliary fauna are essential strategies for the optimisation of biodiversity in different farming systems and pedoclimatic conditions, with the aim of generating resilient and sustainable agroecosystems.

With respect to pest and disease control, integrated indirect measures that include resource conservation mechanisms through biodiversity and the use of preventive methods should be promoted. Direct or curative measures should ultimately be used. Knowledge of the effectiveness of these parallel preventive measures as well as of natural chemical products is essential for success. However, an evaluation of the possible risks that the permitted active substances have on the health of people and the environment is currently increasingly in demand. In addition, weed management requires a transdisciplinary effort to achieve a systemic approach that includes the different tools that are available. The development of more competitive cultivars, the use of cover crops and innovations within physical and mechanical weed management are highlighted. The proper application of available practices increases diversity, reduces the availability of

resources for weeds as well as selection pressure and takes advantage of the positive effects of weeds, all of which are key ecological principles for weed management. In this regard, the disinfection of soil appears to have a great potential for controlling weeds as well as soilborne diseases. However, the great complexity of the chemical, physical and biological processes that regulate this mechanism implies that more research should be pursued on the selection of agro-industrial residues and green manures, quantity of biomass, mode of application and duration of treatment under field conditions that contribute to the non-chemical sanitary defence of vegetable crops. Recent research for biofumigation with plant residues recommends cultural practices according to the type of plant material used. With species of the Brassicaceae family, the chopped aerial part is incorporated into the soil as green manure, increasing its biocidal efficiency if combined with solarisation. Some non-brassicas species show better behaviour in disinfection if we use them as a cover crop in rotation and intercropping.

In summary, more research-specific studies should be carried out in different farming systems and pedoclimatic conditions to achieve the highest efficiency of these established and innovative horticultural practices.

**Author Contributions:** I.d.M.M., D.P.B., J.C., M.E.A., M.M.C., L.O., R.A.R. and J.A.F.; writing—original draft preparation, R.A.R. and J.A.F.; review and editing the manuscript. All authors have read and agreed to the published version of the manuscript.

**Funding:** This research received no external funding.

**Institutional Review Board Statement:** Not applicable.

**Informed Consent Statement:** Not applicable.

**Data Availability Statement:** Not applicable.

**Acknowledgments:** The authors give their thanks to the funding from the Ibero-American Postgraduate University Association (AUIP) for creating the Ibero-American Network for Research in Organic Agriculture (REDIAO).

**Conflicts of Interest:** The authors declare no conflicts of interest.

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
