# Peer review of "Current Trends in Organic Vegetable Crop Production: Practices and Techniques"

_horticulturae, doi:10.3390/horticulturae8100893_

Round 1

Reviewer 1 Report

Line 44:

The diets based on organic products seem to be healthier and tastier, providing a better quality of life for people, compared to diets based on conventional foods [6]. – It is necessary to introduce the issue more deeply. In your source [6] there it is written: The potential role of the production system has not yet adequately been investigated. There is lack of prospective studies and the lack of mechanistic evidence and so, it is presently not possible to determine whether organic food plays a causal role in these observations…. You can of course talk about reducing pesticide residues, dietary patterns or food processing. However, pay an attention that if you will find evidence, that organic fruits is better in vitamins content, it does not automatically mean that it is healthier. This is another scientific approach. Effect of organic farming on food quality is demonstrated by publications from year 2016, 2014 and 2001. This is insufficient. You can find publications that are more recent and actual.

Line 103:

These species/cultivars could be very important in the context of unpredictable climate change conditions and some of them were introduced in the organic market… - please provide some examples.

Line 213:

…other materials such as organic or mineral fertilizers are commonly used – I would recommend to write „…permitted mineral fertilizers…“

Line 224:

Biochard – correctly Biochar

Line 246:

According to reference [70] the bibliographic search was done combining the following fields and logical operators: TITLE(by-product) OR TITLE (waste) AND ALL(biostimulant). This search found 88 articles published in the period 1996–2020. – However, you state something different: period 2015-2021 by searching in Google Scholar for “biostimulants  + organic farming” and “biofertilizers + organic farming”… so who are you referring to?

Line 271:

…although an adequate regulatory framework (for biostimulants and biofertilizers) should be developed. – That has already happened by the new Regulation (EU) no 2019/1009, that provides a clear definition of biostimulants linked to their function. Your biostimulants section should be updated according to this legislation.

Line 352:

Pratilechus penetrans – correctly Pratylenchus penetrans

Line 399 – 408

Brassica and glucosinolates topic is already described in in paragraph 4 and again in the following sections.

Line 496

If you mention flower strips several times, you can also give some specific examples of plant species and potential beneficial arthropods for vegetable production.

Line 568:

Phytium – correctly Pythium

Line 569:

Colletothricum gloesporioides – correctly Colletotrichum gloeosporioides

Line 893:

67. Zhu, J.; Arsovska, B.; Kozovska, K. Acupuncture Treatment in Osteoarthritis. Int. J. Recent Sci. Res. 2020, 11, 37471–37472, 893 doi:10.24327/IJRSR. – how is this reference connected to your topic?

I would appreciate the inclusion of current data from this sector if available (e.g. total area of organic vegetable, produced amount, or % share, the number of available organic cultivars, the number of permitted mineral fertilizers, and plant protection products for organic farming, biostimulants…) otherwise it is, in some parts, just a compilation of general facts.

There is another important issue missing - Greenhouse production in EU’s new legislation for organic production. Please see more: https://ec.europa.eu/commission/presscorner/detail/en/MEMO_17_4686

In addition, what about the issue of the use of copper preparations for plant protection in organic farming?

Article in current stage is not consistent and partly does not cover all aspects of organic vegetable production.

Author Response

Thank you very much for all your comments to the manuscript. They enable us to improve it greatly. The answers to your general comments and particularly questions/suggestions are attached

Reviewer 2 Report

Dear Authors, 

The conclusions could be more summarising, more specific rather than descriptive. They should reflect the tools and methods used on organic farms. In my opinion, the conclusions do not really reflect the current trends in organic vegetable growing.

Author Response

Thank you very much for all your comments to the manuscript. They enable us to improve it greatly. The answer to your general comment is attached

Round 2

Reviewer 1 Report

Dear Authors,

Thank you for your corrections. Now I believe the article contains all the important points.

Please check small mistakes like in the Line 49 – missing bracket or Line 213 there is twice „fertilisers“.